# The Validity of Apple Watch Series 9 and Ultra 2 for Serial Measurements of Heart Rate Variability and Resting Heart Rate

**DOI:** 10.3390/s24196220

**Published:** 2024-09-26

**Authors:** Ben O’Grady, Rory Lambe, Maximus Baldwin, Tara Acheson, Cailbhe Doherty

**Affiliations:** 1School of Public Health, Physiotherapy and Sports Science, University College Dublin, D04V1W8 Dublin, Ireland; ben.ogrady@ucdconnect.ie (B.O.); rory.lambe@ucdconnect.ie (R.L.); maximus.baldwin@ucd.ie (M.B.); tara.acheson@ucdconnect.ie (T.A.); 2Insight SFI Research Centre for Data Analytics, University College Dublin, D04V1W8 Dublin, Ireland; 3Institute for Sport and Health, University College Dublin, D04V1W8 Dublin, Ireland

**Keywords:** heart rate variability, Apple watch, wearable devices, photoplethysmography, validity, Polar H10, Kubios HRV

## Abstract

The widespread use of wearable devices has enabled continuous monitoring of biometric data, including heart rate variability (HRV) and resting heart rate (RHR). However, the validity of these measurements, particularly from consumer devices like Apple Watch, remains underexplored. This study aimed to validate HRV measurements obtained from Apple Watch Series 9 and Ultra 2 against the Polar H10 chest strap paired with the Kubios HRV software, which together served as the reference standard. A prospective cohort of 39 healthy adults provided 316 HRV measurements over a 14-day period. Generalized Estimating Equations were used to assess the difference in HRV between devices, accounting for repeated measures. Apple Watch tended to underestimate HRV by an average of 8.31 ms compared to the Polar H10 (*p* = 0.025), with a mean absolute percentage error (MAPE) of 28.88% and a mean absolute error (MAE) of 20.46 ms. The study found no significant impact of RHR discrepancies on HRV differences (*p* = 0.156), with RHR showing a mean difference of −0.08 bpm, an MAPE of 5.91%, and an MAE of 3.73 bpm. Equivalence testing indicated that the HRV measurements from Apple Watch did not fall within the pre-specified equivalence margin of ±10 ms. Despite accurate RHR measurements, these findings underscore the need for improved HRV algorithms in consumer wearables and caution in interpreting HRV data for clinical or performance monitoring.

## 1. Introduction

The adoption of wrist-worn wearable technology is growing rapidly, with an estimated 31% of the United States population owning a smartwatch [1]. In 2022 alone, over 450 million wearable devices were sold [2]. These devices enable hundreds of millions of users to continuously track biometric data [3]. From basic measurements, such as heart rate and step count, to more sophisticated metrics like sleep quality, energy expenditure, maximal oxygen consumption, peripheral oxygen saturation, and heart rate variability (HRV), these devices have great potential for personal health monitoring [4,5].

HRV is a measure of the variation in the time intervals between successive heartbeats [6]. It is closely associated with the autonomic nervous system, representing the balance between parasympathetic and sympathetic nervous activity [7]. As an indicator of the heart’s response to physiological stress, HRV has been shown to be a useful clinical metric [8]. For example, low and downward-trending HRV has been linked with high stress [9] and increased inflammatory blood markers [10,11,12]. HRV has also been shown to have prognostic value in various medical contexts, including post-myocardial infarction [13,14,15], assessing the risk of sudden cardiac death [16], managing critically ill patients [17], and predicting cancer patient survival [18,19]. Higher HRV values are typically associated with good sleep quality and reflect the increased activity of the parasympathetic nervous system, which promotes relaxation and reduces heart rate [20]. Conversely, low values may be indicative of sleep disruptions, potentially caused by factors including alcohol intake, anxiety, and sleep apnea [21,22].

In athletic populations, HRV can guide training and monitor recovery from exercise by providing insights into an athlete’s autonomic nervous system and overall readiness to perform [23,24,25]. Higher HRV values usually indicate good recovery and resilience, while lower values may signal fatigue or inadequate recovery [26]. Coaches and athletes use HRV to periodize training by adjusting the intensity and volume of workouts based on HRV readings [25]. For instance, lower HRV might suggest reducing training load or focusing on recovery, whereas higher HRV might support more intense or longer sessions. For the general population, HRV can also help guide recreational physical activity planning and monitor overall mental and physical well-being [9].

Findings from this research highlight that a healthy heart is not a metronome; natural variation influenced by heart–brain interactions exists as our bodies adapt to stress stimuli of daily life [27]. The oscillations of a healthy heart are complex and non-linear, requiring advanced mathematical summarizations [28]. Two widely used time-domain metrics for HRV are derived from the QRS complex in electrocardiographic (ECG) recordings: the standard deviation of all normal-to-normal (NN) R-R intervals (SDNN), which measures the variability between heartbeats, and the root mean square of successive differences between normal heartbeats (RMSSD), which assesses the short-term variability in heart rate (HR).

The widespread adoption of smartwatches and fitness trackers by the general public has the potential to significantly enhance personal health monitoring by providing continuous and non-invasive measurement of various biometric data, including HRV. These devices use photoplethysmography (PPG) technology to detect blood volume changes in the wrist, enabling regular HRV assessments without the need for cumbersome equipment [29]. The convenience and accessibility of wearables make them a promising tool for widespread HRV monitoring, which can provide valuable insights into an individual’s autonomic nervous system function and overall well-being. Apple Watch, for instance, uses PPG to calculate HRV through SDNN every two to four hours [29].

However, further research is needed to validate the accuracy of these devices in real-world settings. A recent umbrella review highlighted that fewer than 5% of consumer wearable devices have undergone formal validation for the biometric data they capture [30]. The gold standard for HRV measurement, particularly for assessing cardiac risk, is a 12-lead electrocardiogram (ECG) recorded over a 24-h period, with the results interpreted by an experienced cardiac physician [6,7]. While chest-worn wearables like the Polar H10 (Polar Electro Oy, Kempele, Finland) can accurately record HRV using single-lead ECG [31], they are often impractical for long-term use due to issues related to convenience, comfort, and user compliance.

Therefore, the aim of this study was to evaluate the validity of HRV measurements obtained from Apple Watch Series 9 and Ultra 2 by comparing them with the established reference standard, the Polar H10 chest strap. Specifically, this research sought to assess the accuracy of these devices in measuring HRV under free-living conditions, where factors such as daily life stressors, physical activity, and environmental influences may affect HRV. Additionally, the study aimed to investigate the relationship between resting heart rate (RHR) and HRV measurements, examining whether discrepancies in RHR impact the accuracy of HRV data.

## 2. Materials and Methods

### 2.1. Study Design

This prospective cohort study validated the HRV measurements recorded by Apple Watch Series 9 and Ultra 2. Apple Watch measurements were compared with those obtained from the Polar H10 chest strap paired with the Kubios HRV software, which together served as the reference standard. Ethical approval was granted by the University College Dublin Human Research Ethics Committee (LS-23-55) in 15 November 2023.

### 2.2. Study Population

Participants were recruited through a combination of word of mouth, social media, and posters distributed across relevant locations, including community centers and university campuses. Recruitment materials provided detailed information about the study objectives, procedures, and eligibility criteria to ensure informed participation.

Participants were healthy adults who either owned or were provided with an Apple Watch Series 9 or Ultra 2 for the duration of the study. Upon expressing interest in participation, prospective recruits were informed about the study procedures, and informed written consent was obtained prior to their participation.

### 2.3. Measurement Protocol

After providing informed consent, participants completed an onboarding questionnaire to collect baseline demographic data and information on their Apple Watch usage. They were then familiarized with the use of both Apple Watch and the Polar H10 chest strap, as well as the experimental protocol. Specifically, each morning upon waking, participants were instructed to fit their Apple Watch and Polar H10 according to the manufacturer’s instructions. They simultaneously recorded 5-min HRV readings using both devices. HRV data from the Polar H10 were captured using the Kubios HRV mobile app, while Apple Watch data were recorded via the Breathe app on Apple Watch. Measurements were taken at the same time each morning under consistent, relaxed environmental conditions, with participants lying supine and keeping still. Following the 5-min HRV measurement, participants completed a daily questionnaire (SurveyMonkey Europe UC) and submitted a screenshot of the Kubios HRV app summary page. Participants were asked to follow this protocol daily for between 7 and 14 consecutive days.

Apple Watch Series 9 and Ultra 2 were selected for this study as they were the most recently released models at the time of this trial; they are fitted with Apple’s 3rd generation optical HR sensor (which they share with Apple Watch Series 10). The Polar H10 chest strap was chosen as the reference standard because of its validated accuracy for HRV measurement, with strong agreement and minimal bias compared to 12-lead ECG readings [31]. The Kubios HRV mobile app was utilized for its accessibility and widespread use in HRV analysis, showing excellent reliability and validity in previous studies [32,33].

### 2.4. Statistical Analysis

The primary outcome variable was the difference in HRV, measured by SDNN, between Apple Watch and the Polar H10 paired with the Kubios app. To account for repeated measures (whereby participants contributed multiple observations to the analysis), we employed Generalized Estimating Equations (GEEs) with an autoregressive (AR(1)) working correlation matrix, which assumes that the correlation between repeated measurements decreases as the time between measurements increases. The dependent variable was the difference in HRV between the two devices, and the key covariate was the difference in resting heart rate (RHR) between Apple Watch and the Polar H10. An intercept term was included to estimate the average difference in HRV (SDNN) between the devices.

To assess the accuracy of HRV and RHR measurements, we calculated the mean absolute percentage error (MAPE) and the mean absolute error (MAE) between Apple Watch and the Polar H10. MAPE provided a relative measure of the size of the error in percentage terms, while MAE offered an absolute measure of the differences between the devices. These metrics were used to quantify the degree of accuracy and to compare the performance of Apple Watch with the reference standard.

To further assess the agreement between Apple Watch and the Polar H10, we performed a Bland–Altman analysis; we plotted the difference between the HRV measurements from the two devices against their means, which allowed us to visualize any systematic bias and the limits of agreement (LoAs). The mean difference (bias) and the 95% limits of agreement (calculated as the mean difference ±1.96 times the standard deviation of the differences) were determined.

To determine if the HRV measurements from Apple Watch were equivalent to those from the Polar H10, we also conducted an equivalence test within the GEE framework. The equivalence margin was set at ±10 ms for the difference in HRV, based on prior research and clinical relevance [34,35]. Equivalence was concluded if the 95% Wald confidence interval for the mean difference in HRV fell within these bounds. Model fit was evaluated using the Quasi-Likelihood under Independence Model Criterion (QIC) and the Corrected QIC (QICC). The significance of the covariate (the RHR difference) was assessed using the Wald chi-square test, with a significance level set at *p* < 0.05.

Descriptive statistics were used to summarize and analyze the baseline characteristics of the study participants, including demographic information, such as age, gender, and body mass index (BMI). Means and standard deviations (SDs) were calculated for continuous variables, while frequencies and percentages were used for categorical variables.

All analyses were performed using SPSS version 29.

## 3. Results

### 3.1. Participant Demographics

A total of 39 participants were included in the study, contributing 316 HRV measurements. The mean age of the participants was 24.6 years (SD = 8.2), with 17 males and 22 females. The mean BMI was 23.65 kg/m^2^ (SD = 1.97), the mean height was 1.69 m (SD = 0.34), and the mean weight was 73.08 kg (SD = 10.52).

### 3.2. Comparison of HRV and Resting Heart Rate Measurements

The mean difference in HRV (SDNN) between Apple Watch and the reference standard was −8.31 ms (SD = 20.46 ms), with a 95% confidence interval ranging from −11.04 ms to −5.59 ms. The mean difference in resting heart rate between Apple Watch and the reference standard was 0.08 bpm (SD = 3.73 bpm), with a 95% confidence interval ranging from −0.78 bpm to 0.93 bpm.

The MAPE and MAE were calculated to assess the accuracy of Apple Watch compared to the reference standard. For HRV (SDNN), the MAPE was 28.88% (95% CI: 26.18% to 31.57%) and the MAE was 20.46 ms (95% CI: 18.57 to 22.34 ms). For resting heart rate, the MAPE was 5.91% (95% CI: 4.78% to 7.03%) and the MAE was 3.73 bpm (95% CI: 2.97 to 4.49 bpm).

These results are summarized in Table 1.

#### 3.2.1. Bland–Altman Analysis

To further assess the agreement between Apple Watch and the reference standard (Polar H10 and Kubios) for HRV (SDNN) measurements, a Bland–Altman plot was generated. The plot visually represented the mean difference (bias) and the limits of agreement (LoAs) between the two devices. This is displayed in Figure 1.

#### 3.2.2. Generalized Estimating Equations

The GEE model revealed a significant intercept for HRV (B = −8.31 ms, 95% CI: [−11.04, −5.59], *p* = 0.025), indicating that, on average, Apple Watch underestimated HRV (SDNN) by 8.31 ms compared to the reference standard. The effect of differences in resting heart rate on HRV differences was not statistically significant (B = 0.36 ms per bpm, 95% CI: [−0.14, 0.85], *p* = 0.156), suggesting that variations in resting heart rate did not significantly impact HRV measurement differences.

#### 3.2.3. Equivalence Testing

The equivalence test aimed to determine if the HRV measurements from Apple Watch were equivalent to those from the Polar H10 and Kubios software (version: 1.5.0) within a pre-specified margin of ±10 ms. The 95% confidence interval for the mean difference in HRV (SDNN) was −11.04 ms to −5.59 ms. Since this confidence interval extended beyond the upper bound of the equivalence margin (10 ms), equivalence could not be concluded. This result suggests that while Apple Watch’s HRV measurements were generally close to those of the reference standard, the observed differences exceeded the acceptable equivalence margin, indicating that Apple Watch’s HRV measurements cannot be considered equivalent to the reference standard within a meaningful range of ±10 ms.

These findings are summarized in Figure 2.

## 4. Discussion

This study validated HRV measurements from Apple Watch Series 9 and Ultra 2 against those recorded by the Polar H10 chest strap paired with the Kubios HRV software, which served as the reference standard. Our findings indicate that Apple Watch tends to underestimate HRV (measured using SDNN) by an average of 8.31 ms. Variations in resting heart rate (RHR) measurements between Apple Watch and the reference standard did not significantly impact HRV measurement differences, suggesting that the inaccuracies observed in HRV readings may not be directly related to errors in the PPG-derived heart rate measurements. Overall, the RHR measurements demonstrated close levels of agreement (mean difference: −0.1 bpm).

The discrepancies in HRV measurement were consistent with previous research, which also found a tendency for consumer wearables to underestimate values [36,37,38]. For instance, a validation study of six wearable devices found that Apple Watch Series 6 underestimated HRV by an average of 9.6 ms [38]. Although these authors invited the device manufacturers to provide raw data to enhance the precision of the comparisons, not all accepted, and this may have led to unfavorable comparisons between certain devices. Raw data were also used for HRV calculations in a separate study [37], and while this approach provided an explicit assessment of the RR intervals captured by Apple Watch, it did not directly evaluate the native HRV values from the device, as the proprietary software was not used for their calculation. This distinction emphasizes the need for validation studies that compare not only raw data outputs but also the performance of algorithms used by consumer wearables which generate the values presented to end users.

For this reason, we considered it important to control for the accuracy of the underlying RHR readings in our GEE analysis. While the effect of differences in RHR on HRV differences was not statistically significant and a low mean difference was found, the mean absolute and mean average percentage errors were larger. This variance may be indicative of outliers among the data or larger deviations in certain individual comparisons. Although inaccurate reporting of measurements by participants is one potential source of this deviation, even small discrepancies in the RHR measurements recorded by Apple Watch could potentially impact HRV readings, given that the metric relies on millisecond-level accuracy. However, the strong overall agreement between RHR measurements reinforces the reliability of the device in capturing heart rate data.

To determine the utility of Apple Watch’s readings, given the observed discrepancies, the clinical relevance of its short-term SDNN HRV measurements requires further consideration. SDNN is regarded as the gold standard for medical stratification of cardiac risk when measured over a 24-h period [7]. LF band power makes a significant contribution to such recordings, and these longer measurements provide data about cardiac reactions to a greater range of environmental stimuli when compared to shorter-term measurements [6]. Based on 24-h recording, patients with SDNN values over 100 ms demonstrate a lower risk of mortality, whereas individuals with values below 50 ms may have compromised health [39]. However, resting values obtained from short-duration measurement correlate poorly with 24-h indices, and their physiological meanings may differ [40]. Although small changes in HRV within the range of 5−10 ms may be meaningful in specific clinical contexts, such differences may be less impactful for general health monitoring, and given the marked inter-individual variation that exists for HRV measurement [41], it is difficult to establish specific thresholds for clinically important changes. The practical implication is that while Apple Watch may provide valuable insights for trend analysis or relative changes in HRV over time, caution should be exercised when interpreting absolute HRV values for clinical decision making. Whether short-term SDNN is the most appropriate measure of HRV, particularly when assessed using consumer wearables like the Apple Watch (compared to other metrics such as RMSSD), remains a topic of ongoing debate. Further research is needed to fully determine its reliability and relevance across both clinical and non-clinical settings.

One of the primary challenges in wearable technology research is conducting agile validation studies that can keep pace with the rapidly evolving commercial ecosystem and the annual release cycle of devices like Apple Watch. This is exemplified by the findings of a recent umbrella review highlighting that most consumer wearables have been replaced by newer models by the time validation research assessing their accuracy is published, and that <5% of devices released to data have been validated [30]. Our protocol was designed with this challenge in mind, emphasizing the practical application of HRV measurement in free-living conditions over a two-week period. This approach contrasts with the more controlled environments of laboratory-based HRV validation studies, which may not fully capture the variability and real-world conditions in which these devices are typically used. Home-based morning measurements offer several advantages. They are less burdensome for participants, replicating the everyday context in which wearable data are most often collected, thereby enhancing the ecological validity of our findings. This practical and accessible protocol also makes it easier to implement and ensures that measurements are taken in an idealized at “rest” physiological state, reducing potential biases introduced by the stress or discomfort that can accompany laboratory visits [42,43]. By capturing HRV data in a naturalistic setting, we designed this study to reflect the true utility and performance of wearable devices like Apple Watch in daily life while also addressing the need for more flexible and responsive research methodologies in this rapidly advancing field.

The benefits of this approach are important, but they are accompanied by several limitations. First, while opportunistic, real-world data capture is less burdensome for participants, it is also unsupervised and uncontrolled; this ‘citizen science’ approach has the potential to introduce error and deviation from the intended protocol in the absence of oversight from the research team. To tackle this, all participants were thoroughly familiarized with the protocol in advance of data collection and were sent regular reminders to complete the protocol each morning. Second, the two-week protocol may not fully capture long-term variations in HRV or changes across different contexts, such as varying levels of physical activity, stress, or a complete menstrual cycle. Lastly, a relatively small convenience sample consisting of predominantly young, healthy adults was recruited. This limits the extrapolation of our findings to older adults or those with varying health conditions. Studies with more diverse populations would facilitate an improved understanding of Apple Watch’s HRV measurement accuracy.

## 5. Conclusions

This study demonstrated that while Apple Watch Series 9 and Ultra 2 tend to underestimate HRV compared to the Polar H10 chest strap, a well-established reference standard, they nonetheless provide a relatively accurate measure of resting heart rate. Despite the observed discrepancies, which suggest that further refinement is needed to achieve clinical-level accuracy in HRV measurements, our study’s ecological validity and robust statistical approach offer valuable insights into the potential and current limitations of consumer-grade wearables for HRV monitoring. Future research should focus on improving data accuracy, extending data collection periods, and including more diverse populations to enhance the generalizability and clinical applicability of HRV data obtained from wearable devices. As wearable technology continues to evolve, these improvements will help to bridge the gap between consumer-grade devices and clinical standards, ultimately enhancing their utility in both personal health management and clinical practice.

## Figures and Tables

**Figure 1 sensors-24-06220-f001:**
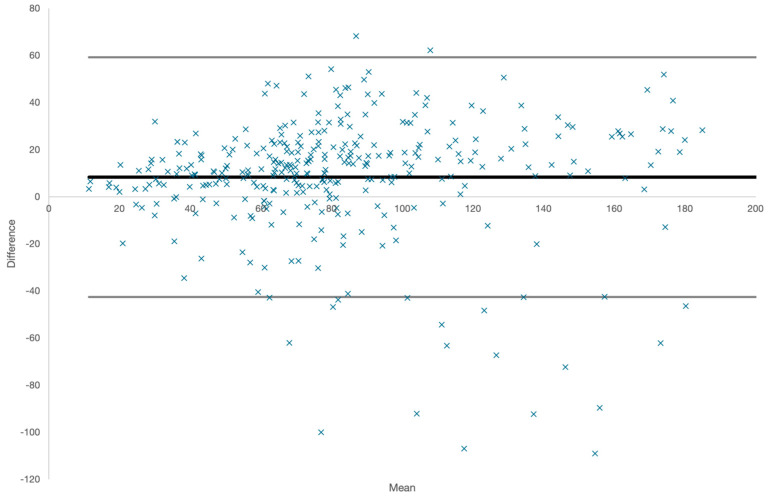
Bland–Altman plot for HRV (SDNN) measurements ^1^. ^1^ The *x*-axis represents the means of the HRV values measured by both devices, while the *y*-axis shows the differences between the HRV measurements (Apple Watch minus Polar H10). The solid black line represents the mean difference (bias) of −8.31 ms, indicating that Apple Watch generally underestimates HRV compared to the Polar H10. The dashed lines indicate the 95% limits of agreement (LoAs), calculated as the mean difference ±1.96 times the standard deviation of the differences, which range from −53.8 ms to 37.2 ms.

**Figure 2 sensors-24-06220-f002:**
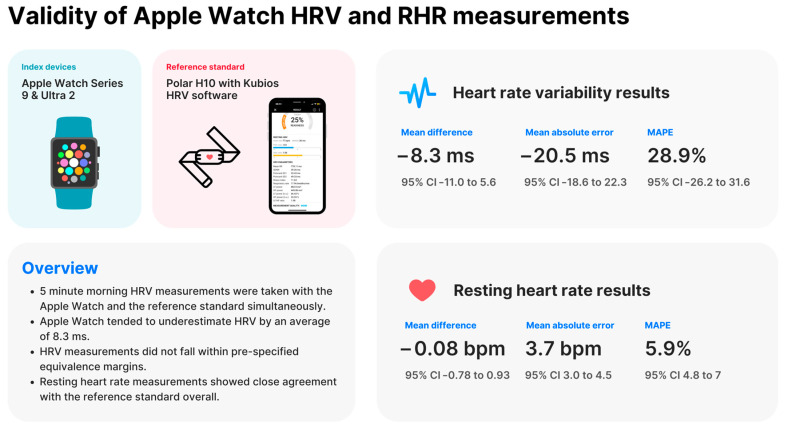
Summary of results. bpm = beats per minute, HRV = heart rate variability, MAPE = mean absolute percentage error, RHR = resting heart rate.

**Table 1 sensors-24-06220-t001:** Comparison of mean HRV and resting heart rate measurements between Apple Watch and Polar H10 + Kubios reference standard.

Measurement	Kubios	Apple Watch	Mean Difference	MAE	MAPE
HRV (SDNN)	85 ms	93.3 ms	−8.3 ms (95% CI: −11 to −5.6)	20.5 ms (95% CI: 18.6 to 22.3)	28.9% (95% CI: 26.2% to 31.6%)
Resting Heart Rate	60.7 bpm	60.7 bpm	−0.08 bpm (95% CI: −0.78 to 0.93)	3.7 bpm (95% CI: 3 to 4.5)	5.9% (95% CI: 4.8% to 7%)

bpm = beats per minute, HRV = heart rate variability, MAE = mean absolute error, MAPE = mean absolute percentage error, SDNN = standard deviation of NN intervals.

## Data Availability

The data supporting the reported results of this study are available on OSF at (https://osf.io/cf453/, (accessed on 31 July 2024)).

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
