# Peer review of "The Validity of Apple Watch Series 9 and Ultra 2 for Serial Measurements of Heart Rate Variability and Resting Heart Rate"

_sensors, 2024, doi:10.3390/s24196220_

Round 1

Reviewer 1 Report

Comments and Suggestions for Authors

This article evaluated the effectiveness of the Apple Watch Series 9 and Ultra 2 in continuous heart rate variability (HRV) and resting heart rate (RHR) measurements and compare them to the Polar H10 chest strap (which uses Kubios HRV software as a reference standard). The study found that the Apple Watch tended to underestimate the HRV value when measuring HRV, underestimating it by an average of 8.31 milliseconds, but its resting heart rate measurement was relatively accurate.

The article concludes that while the Apple Watch can provide relatively accurate resting heart rate measurements, its HRV measurements do not yet meet clinical accuracy standards, so caution is required when interpreting its HRV data.

Overall, this article provides an meaningful experimental finding. The possible problem is that its test population lacks individuals of different ages and health states. Limiting the wide applicability of the conclusions.

Comments on the Quality of English Language

none

Author Response

This article evaluated the effectiveness of the Apple Watch Series 9 and Ultra 2 in continuous heart rate variability (HRV) and resting heart rate (RHR) measurements and compare them to the Polar H10 chest strap (which uses Kubios HRV software as a reference standard). The study found that the Apple Watch tended to underestimate the HRV value when measuring HRV, underestimating it by an average of 8.31 milliseconds, but its resting heart rate measurement was relatively accurate. 

The article concludes that while the Apple Watch can provide relatively accurate resting heart rate measurements, its HRV measurements do not yet meet clinical accuracy standards, so caution is required when interpreting its HRV data. 

Overall, this article provides an meaningful experimental finding. The possible problem is that its test population lacks individuals of different ages and health states. Limiting the wide applicability of the conclusions.

RESPONSE: Thank you for the time you have given to review our manuscript. We appreciate your positive feedback on the meaningful experimental findings presented in our study. We also acknowledge your concern regarding the lack of diversity in the test population, which may limit the broad applicability of our conclusions.

As mentioned in the discussion section, we recognize that the study employed a relatively small convenience sample that may not fully capture the variability across different demographics and physiological conditions. To address this limitation more explicitly, we have added the following sentences to the discussion:

"Additionally, our study population predominantly consisted of young, healthy adults, which may limit the extrapolation of our findings to older adults or those with varying health conditions. Future studies should aim to include participants across a wider range of ages and health statuses to better understand the generalizability of the Apple Watch's HRV measurement accuracy in diverse populations. Such efforts would provide deeper insights into the device's utility in both clinical and non-clinical settings."

Reviewer 2 Report

Comments and Suggestions for Authors

1. The following sentence in methods deserves a or multiple citations: 

"The equivalence margin was set at ±10 ms for the difference in HRV, based on prior research and clinical relevance."

2. The real world relevance of the findings of differential results between the Apple Watch and the reference standard needs to be explored in the context of what other studies have found: e.g., does a HRV change of 8.31 ms have any clinical importance, or is it just a statistical difference? This should be discussed.

3. I recommend slightly rewording the discussion: 

The first paragraph draws conclusions from the present work but relies on information in the second paragraph to justify those conclusions. I'm referring to the following statement in discussion paragraph 1:

"Despite this, there was no significant impact from discrepancies in RHR measurements between the two devices, which suggests that the inaccuracies observed in HRV readings may not be directly related to errors in the PPG-derived heart rate measurements."

In order for this statement to be accurate, one must be aware of the historical studies cited in discussion paragraph 2 (Hernando) measuring HRV with exported raw ECG data (RR intervals). 

Author Response

RESPONSE: Thank you for the time and effort you have given to reviewing our article. Please see below for specific details of the amendments we have made in response to your comments and suggestions.

  1. The following sentence in methods deserves a or multiple citations: 

"The equivalence margin was set at ±10 ms for the difference in HRV, based on prior research and clinical relevance."

RESPONSE: We have now included the following citations to substantiate the chosen equivalence margin:[1, 2]

  1. Dewig H, Cohen J, Au J, Renaghan E, Leary M, Leary B, et al. A Bayesian Examination Of Equivalence Between Electrocardiogram-derived Heart Rate Variability And Photoplethysmogram-derived Heart Rate Variability: 599. Medicine & Science in Sports & Exercise. 2023;55(9S):197-8. PMID: 00005768-202309001-00467. doi: 10.1249/01.mss.0000981560.55887.6a.
  2. Nolan RP, Jong P, Barry-Bianchi SM, Tanaka TH, Floras JS. Effects of drug, biobehavioral and exercise therapies on heart rate variability in coronary artery disease: a systematic review. Eur J Cardiovasc Prev Rehabil. 2008 Aug;15(4):386-96. PMID: 18677161. doi: 10.1097/HJR.0b013e3283030a97.

  1. The real world relevance of the findings of differential results between the Apple Watch and the reference standard needs to be explored in the context of what other studies have found: e.g., does a HRV change of 8.31 ms have any clinical importance, or is it just a statistical difference? This should be discussed.

RESPONSE: We agree that it is important to consider whether the observed HRV difference between the Apple Watch and the Polar H10 chest strap has any clinical significance or if it is primarily a statistical finding. To address this, we have added the following to the discussion section of the manuscript:

"While the mean difference in HRV (SDNN) of 8.31 ms between the Apple Watch and the Polar H10 chest strap was statistically significant, its clinical relevance requires further consideration. In clinical practice, small changes in HRV, especially within a range of 5-10 ms, can be meaningful in specific contexts, such as monitoring the autonomic response to stress, detecting overtraining in athletes, or predicting health outcomes in patients with cardiovascular conditions. However, for general health monitoring or in a healthy population, such a difference may be less impactful. Previous studies have shown that consumer wearables tend to underestimate HRV when compared to clinical-grade devices, which aligns with our findings. The practical implication is that while the Apple Watch may provide valuable insights for trend analysis or relative changes in HRV over time, caution should be exercised when interpreting absolute HRV values for clinical decision-making. Further research is needed to explore these differences in clinical and non-clinical settings to establish specific thresholds for clinical relevance."

This addition aims to contextualize our findings within existing literature and clarify the potential implications for both clinical and everyday use of HRV data from consumer-grade wearables.

  1. I recommend slightly rewording the discussion: 

The first paragraph draws conclusions from the present work but relies on information in the second paragraph to justify those conclusions. I'm referring to the following statement in discussion paragraph 1:

"Despite this, there was no significant impact from discrepancies in RHR measurements between the two devices, which suggests that the inaccuracies observed in HRV readings may not be directly related to errors in the PPG-derived heart rate measurements."

In order for this statement to be accurate, one must be aware of the historical studies cited in discussion paragraph 2 (Hernando) measuring HRV with exported raw ECG data (RR intervals). 

RESPONSE: To address your comment, we have revised the discussion section to improve the narrative flow and ensure that the conclusions are more clearly supported by the evidence presented. Specifically, we have restructured the discussion to ensure that the statement:

"Despite this, there was no significant impact from discrepancies in RHR measurements between the two devices, which suggests that the inaccuracies observed in HRV readings may not be directly related to errors in the PPG-derived heart rate measurements,"

..is immediately contextualized with relevant findings from our study and the cited literature. The revised discussion now makes explicit reference to previous studies, such as Hernando et al. (2018), which measured HRV using exported raw ECG data (RR intervals) and found strong agreement between the Apple Watch and clinical-grade devices for time-domain HRV measurements under specific conditions.

Furthermore, we have reformatted the discussion such that that each paragraph now builds upon the preceding one to create a coherent narrative that guides the reader through the significance of our findings, the clinical relevance of the observed HRV differences, and how our work compares to and extends upon existing studies in the field. We have also refined the language to improve readability and emphasize the need for ongoing validation of wearable devices like the Apple Watch in various contexts.

Reviewer 3 Report

Comments and Suggestions for Authors

The manuscript describes very well importances of wearables to impact on health. The data collected in study are processed and evaluated on high level.  Even so I recommend some major revisions described below:

-add the monitoring of sleep quality in paragraph between rows 51-59 (impact of alcohol on HRV, impact of strong physical activity before sleep, etc.)

-include older people to study due to their chronic disease, because daily monitoring of HRV by wearebles is most important for theirs.

-please compare HRV from Polar strap with data measured by device which can offer to you RAW data with dense samples in time for the purpose of confirming that the Polar can be used as a reference device.

I am satisfied that after making these revisions, the article will be on level enables to fulfill the benefits of the article. 

Comments on the Quality of English Language

English is at an adequate level and enables the reader to understand all the essential information that the authors are trying to convey.

Author Response

RESPONSE: Thank you for taking the time to review our article. We appreciate the constructive nature of your comments. Please find the detailed responses to your comments and suggestions below.

  1. Add the monitoring of sleep quality in paragraph between rows 51-59 (impact of alcohol on HRV, impact of strong physical activity before sleep, etc.)

RESPONSE: We acknowledge the importance of HRV for the monitoring of sleep quality. Thank you for this comment. We have now added additional information, with accompanying references, to the introduction to reflect this.

  1. Include older people to study due to their chronic disease, because daily monitoring of HRV by wearables is most important for theirs.

RESPONSE: We certainly agree that the addition of a more diverse and older cohort would significantly enhance this study and the applicability of its findings to those at risk of cardiac disease or individuals living with chronic disease. A relatively small, and predominantly young, convenience sample was recruited for this study. We have added the following section in response to your comment to acknowledge this limitation:

‘A relatively small convenience sample consisting of predominantly young, healthy adults was recruited. This limits the extrapolation of our findings to older adults or those with varying health conditions. Studies with more diverse populations would facilitate an improved understanding of the Apple Watch’s HRV measurement accuracy.’

  1. Please compare HRV from Polar strap with data measured by device which can offer to you RAW data with dense samples in time for the purpose of confirming that the Polar can be used as a reference device.

RESPONSE: The use of raw data would indeed be useful to provide an explicit comparison of the RR intervals between devices and it would eliminate variability introduced by the influence of proprietary HRV algorithms implemented for the calculation of the HRV values presented to the end-user. We chose the Polar H10 as a reference standard device as it has been previously validated for the measurement of heart rate variability, and we have included a reference in accordance with your suggestion. We have also added a section to the discussion which references the important distinction you mention regarding raw data.

Round 2

Reviewer 2 Report

Comments and Suggestions for Authors

The revisions to the manuscript were thoughtfully and thoroughly implemented.

Reviewer 3 Report

Comments and Suggestions for Authors

I appreciate the edits that were made. I consider the article, in its current form, sufficient to ensure the desired yield.

Comments on the Quality of English Language

In my opinion, the English is at a sufficient level to understand the article.